

# Retrospective analysis of estrogen receptor (ER), progesterone receptor (PR), human epidermal growth factor receptor-2 (HER2), Ki67 changes and their clinical significance between primary breast cancer and metastatic tumors

Gaoxiu Qi, Xin Zhang, Xiaoying Gai and Xiong Yan

Qingdao Central Hospital, University of Health and Rehabilitation Sciences (Qingdao Central Medical Group), Qingdao, Shandong, China

Corresponding author
Xiong Yan, 181825012@qq.com

## ABSTRACT

**Objective**. To explore the relationship between receptor heterogeneity and clinicopathological characteristics in 166 patients with invasive breast cancer during metastasis.

**Methods**. We conducted a retrospective analysis of 166 patients diagnosed with metastatic breast cancer through biopsy, who were admitted to our hospital from January 2018 to December 2022. Statistical analysis was employed to assess the heterogeneity of receptors in both primary and metastatic lesions, including estrogen receptor (ER), progesterone receptor (PR), human epidermal growth factor receptor-2 (HER2), Ki67, as well as their association with clinicopathological features such as tumor size, lymph node metastasis, treatment regimen, and disease-free survival.

**Results**. The discordant expression rates of ER, PR, HER2, Ki-67 and Luminal classification between primary and metastatic lesions were 21.7%, 41.6%, 8.9%, 34.4% and 36.8%, respectively. There is a significant difference in disease-free survival between patients with consistent and inconsistent receptor status of primary and metastatic lesions, which is statistically significant. The median DFS for primary HER2(-) to metastatic HER2(+) was 84 months, which was relatively high. The Cox multivariate regression analysis revealed that the expression differences of ER, PR, HER2, and Ki67 were not influenced by endocrine therapy and chemotherapy. However, a statistically significant difference in HER2 expression was observed with targeted therapy. Tumor size was correlated with ER and Ki67 receptor status ($P = 0.019, 0.016$). Tumor size was not correlated with PR, and HER2 ($P = 0.679, 0.440$). Lymph node metastasis was not associated with changes in ER, PR, HER2, and Ki67. The discordant rates of ER, PR, HER2, and Ki-67 in patients with local recurrence were 22%, 23.7%, 5.1%, and 28.8% respectively, whereas those in patients with distant metastasis were 21.5%, 36.4%, 10.3%, and 31.8% respectively.

**Conclusions**. The expression levels of ER, PR, HER2, and Ki-67 in primary and metastatic breast cancer exhibit heterogeneity, which is closely associated with the prognosis and treatment outcomes of patients.

# INTRODUCTION

The incidence and mortality rates of breast cancer are the highest among all female malignant tumors, causing significant harm to women's health (*Sung et al., 2021*). The expression of estrogen receptor (ER), progesterone receptor (PR), human epidermal growth factor receptor 2 (HER-2), and Ki67 is commonly employed in clinical practice to categorize breast cancer subtypes, guiding the selection of appropriate treatment methods. The treatment and prognosis of breast cancer patients are closely correlated with the expression of receptors in the primary lesions. Although the majority of breast cancer patients undergo surgical, endocrine, or chemotherapy interventions, there is still a subset comprising 20–30% who experience metastasis to lymph nodes, chest wall, bone, or liver (*Dieci et al., 2013*; *Mellouli et al., 2022*; *Schrijver et al., 2018*). There is increasing evidence indicating that approximately 31% of primary and metastatic tumors exhibit altered receptor expression, which necessitates modifications to the treatment plan (*Broom et al., 2009*; *Kao et al., 2021*; *Liedtke et al., 2009*). Inconsistency of receptors can affect prognosis, and the absence of receptors is associated with poor prognosis, as well as different stages, treatment plans, and metastatic sites of the breast (*Shiino et al., 2022*). The present study aims to provide additional clinical references by conducting a retrospective analysis of 166 cases of invasive breast cancer patients, focusing on the heterogeneity in ER, PR, and HER2 expression between primary tumors and metastases, as well as the impact of this variation on prognosis and individualized treatment.

# MATERIAL AND METHODS

## Clinical data

The clinicopathological data of 166 patients with biopsy-confirmed invasive breast cancer metastasis at Qingdao Central Hospital Affiliated to Qingdao University from January 2018 to November 2023 were retrospectively analyzed. The analysis included the following variables: gender, age, primary site (left or right), histological type and grade, date of first operation, endocrine therapy, chemotherapy, targeted therapy, EP, PR, HER2 and Ki67 values of the primary site; date and site of metastasis, EP, PR, HER2 and Ki67 of the metastatic lesion. The patient mainly provides treatment plans based on the guidelines of the National Comprehensive Cancer Network (NCCN) (*Gradishar et al., 2023*) in the United States, combined with the patient's actual situation. Inclusion criteria: female patients diagnosed with unilateral primary or metastatic breast cancer, who are eligible for surgery or biopsy; complete clinical and follow-up data were collected. Exclusion criteria: male patients with breast cancer; patients with bilateral breast cancer; patients with a history of non-breast malignant tumors; patients with incomplete clinical and follow-up data. The Ethics Committee of Qingdao Central Medical Group approved this retrospective study (approval number: KY202304801), and that no participant consent was obtained.

## Production process of immunohistochemistry

The surgeon will immediately cut open the surgical specimen along the maximum diameter of the tumor after it is removed from the body. The surgeon will immediately add 10% neutral formalin fixative, which is 5-10 times the amount of tissue. The fixation time for the primary lesion specimen should not exceed 30 min after removal, and for the puncture specimen, it should not exceed 10 min after removal. We used fully automated Roche staining for all immunohistochemistry methods, including ER (monoclone rabbit antibody), PR (monoclone rabbit antibody), HER2 (monoclone rabbit antibody), Ki67 (MIB1; monoclone mouse antibody), all purchased from Ventura Medical Systems. The thickness of paraffin embedded tissue slices is 3 µm, and after baking in a 60 °C for 2 h, they are subjected to automatic Roche staining and manually sealed. The pathologist will observe under an optical microscope (OLYMPUS) after staining. As for ER, PR, and Ki67 immunohistochemistry, we used internal and external controls, while we used residual normal mammary ductal epithelial cells in the tissue as internal controls. As for HER2, we use an external control, and each slice will have quality control tissue. Only when the external control slice is positive, we will interpret HER2 on the target slice. Otherwise, we will perform immunohistochemical staining again. All pathological results were evaluated by two senior pathologists for diagnosis. When breast cancer is diagnosed in other institutions, unstained slides are retrieved through referral institutions and re analyzed in the pathology department of our hospital.

## Criteria for the interpretation of immunohistochemical staining

The staining of ER and PR was characterized by the presence of yellow or brown granules in the nucleus. A cell count $\geq 1\%$ indicated a positive result, while $<1\%$ indicated a negative result. A PR positive rate $\geq 30\%$ was considered high expression, whereas a PR positive rate $<30\%$ was classified as low expression. HER2 grouping criteria were as follows: HER2 (0), HER2 (1+) and HER2 (2+) without amplification by DISH/FISH were categorized as HER2 negative group; HER2 (2+) with DISH/FISH amplification and HER2 (3+) were classified as HER2 positive group. The Ki67 interpretation criteria involved selecting and counting three or more hot spots of invasive cancer positive cells in high-power fields to determine the average Ki67 index. A Ki67 expression level $\geq 30\%$ indicated high expression, while $<30\%$ indicated low expression.

## Follow up on the site of metastasis and disease-free survival time of patients

The follow-up period commenced on the day of the initial surgical procedure and continued until the occurrence of metastases, encompassing both local recurrence and distant metastasis. Local recurrence comprised ipsilateral breast, ipsilateral chest wall, and ipsilateral regional lymph node metastases. Distant metastases included contralateral breast, contralateral chest wall, contralateral lymph nodes, bilateral supraclavicular lymph nodes, bone involvement, as well as visceral spread. The follow-up process involved a combination of outpatient reexamination, in-patient reexamination, and telephone follow-up. Disease-free survival (DFS) was defined as the interval between the first operation and the onset of initial metastasis.

## Statistical methods

The data analysis and graph generation were performed using SPSS 23.0 and GraphPad Prism 8.0.2 statistical software. Descriptive statistics were utilized to summarize the clinicopathological data of patients diagnosed with primary breast cancer and metastatic breast cancer, with the results presented as median values, case numbers, and percentages. The Kappa consistency test was employed to assess the concordance of ER/PR/HER2/Ki67 between primary and metastatic lesions. The Kaplan–Meier method was employed for survival analysis, and survival curves were generated. The Log-rank test was utilized to compare the differences in ER, PR, HER2, and Ki67 between the consistent group and the inconsistent group in both primary and metastatic lesions. A Cox regression model was employed for conducting multivariate analysis of treatment regimen. The chi-square test was employed to examine the association between recipient status and tumor size or lymph node metastasis. The criterion for statistical significance was set at a level of $P < 0.05$.

## RESULT

### Clinicopathological features

A total of 166 patients met the inclusion criteria. The age range of the patients was 25 to 78 years, with a median age of 51 years. All patients were female. There were 85 cases of left breast cancer and 81 cases of right breast cancer. The pathological diagnosis for all patients was invasive carcinoma, including up to 156 cases of ductal carcinoma. Among them, 76 cases (45.8%) were classified as ESBR grade II. Complete T stage information was available for 154 patients and complete N stage information was available for 160 patients. 85 patients received endocrine therapy, while chemotherapy and targeted therapy were administered to 160 and 34 patients respectively. All 166 patients with breast cancer had metastasis, including 43 with local recurrence and 123 with distant metastasis (Table 1).

### Heterogeneity of receptor expression in primary and metastatic lesions

A total of 166, 165, 156, and 146 patients had complete ER, PR, HER2, and Ki-67 data for primary and metastatic lesions, respectively. The rate of positive staining for ER, PR, HER2 in primary lesions was 66.9%, 59.4%, 26.3%, respectively, while that in metastatic lesions was 57.2%, 41.8%, 28.2%, respectively. The rate of high expression of Ki-67 was 67.1% in primary lesions, while that in metastatic lesions was 69.9% (Table 2). In Table 2, the Kappa values of ER, PR, HER2, Ki67 expression and molecular subtyping between primary and metastatic lesions were 0.569, 0.377, 0.790, 0.175 and 0.493, respectively. The concordance of HER2 expression between primary and metastatic lesions was high. On the other hand, the rate of inconsistent expression for ER, PR, HER2, and Ki-67 was 21.7%, 32.1%, 8.3% and 35.6%, respectively, with an average of 24.4%, between primary and metastatic lesions. According to the expression of ER, PR, HER2 and Ki67, all breast cancer patients were classified into molecular classification (Table 3), among which 12 cases of Luminal B(HER2-) became Basal type, accounting for the highest proportion. The details of the change types and results are in Table 3.

In addition, the following presentation shows HE, ER, PR, HER2, and Ki67 in the primary and metastatic lesions of three cases with molecular typing changes. The histological grades

**Table 1** The clinicopathological characteristics of 166 patients with breast cancer were analyzed descriptively (*n* (%)).

| Characteristic | | Primary lesion | Metastatic lesion |
|---|---|---|---|
| Median age, years | Median (Range) | 51(25–78) | 54(28–80) |
| | ≤35 | 17(10.2%) | 7(4.2%) |
| | >35 | 149(89.8%) | 159(95.8%) |
| Gender | Female | 166(100%) | 166(100%) |
| Orientation | Left | 85(51.2%) | Not applicable |
| | Right | 81(48.8%) | |
| Pathology | Ductal | 156(94.0%) | Not applicable |
| | Other | 10(6.0%) | |
| Grade | One | 3(1.8%) | Not applicable |
| | Two | 83(50.0%) | |
| | Three | 49(29.5%) | |
| | Unknown | 31(18.7%) | |
| T | I | 54(32.5%) | Not applicable |
| | II | 76(45.8%) | |
| | III | 24(14.5%) | |
| | Unknown | 12(7.2%) | |
| N | N0 | 32(19.3%) | Not applicable |
| | N1 | 63(38.0%) | |
| | N2 | 36(21.7%) | |
| | N3 | 29(17.4%) | |
| | Unknown | 6(3.6%) | |
| Endocrine therapy | Yes | 85(51.2%) | Not applicable |
| | No | 81(48.8%) | |
| Chemotherapy | Yes | 160(96.4%) | Not applicable |
| | No | 6(3.6%) | |
| Targeted Therapy | Yes | 34(20.5%) | Not applicable |
| | No | 132(79.5%) | |
| Site of metastasis | Local recurrence | Not applicable | 43(25.9%) |
| | Distant metastasis | | 123(74.1%) |

of the primary lesion in the three cases were II, III, and III. The maximum diameters of the primary lesion were 1.6 cm, 2.3 cm, and 2.5 cm, respectively. The N stages were N1, N1, and N2, and the disease-free survival periods were 34, 14, and 24 months, respectively. The metastatic lesions were neck, liver, and liver. The top two cases only underwent endocrine and chemotherapy. The third case only received chemotherapy. The final molecular typing changes of the three cases were as follows: primary lesion Luminal A transformed into metastatic lesion HER2 overexpression, primary lesion Luminal B (HER2-) transformed into metastatic lesion Luminal A, and primary lesion Basal type transformed into metastatic lesion Luminal B (HER2-). The detailed changes in HE, ER, PR, HER2, and Ki67 images are shown in Fig. 1.

**Table 2 Descriptive analysis and Kappa consistency test were used to determine the expression of ER, PR, HER2, Ki67 receptor status, and molecular subtyping in primary and metastatic lesions after excluding unknown data ($n$ (%)).**

| Receptor | Receptor status | Primary lesion | Metastatic lesion | Kappa |
|---|---|---|---|---|
| ER (166 cases) | Positive (+) | 111(66.9%) | 95(57.2%) | 0.569 |
| | Negative (−) | 55(33.1%) | 71(42.8%) | |
| PR (165 cases) | Positive (+) | 98(59.4%) | 69(41.8%) | 0.377 |
| | Negative (−) | 67(40.6%) | 96(58.2%) | |
| HER2 (156 cases) | Positive (+) | 41 (26.3%) | 44(28.2%) | 0.790 |
| | Negative (−) | 115 (73.7%) | 112(71.8%) | |
| Ki67 (146 cases) | High expression | 98(67.1%) | 102(69.9%) | 0.175 |
| | Low expression | 48 (32.9%) | 44(30.1%) | |
| Molecular subtyping (151 cases) | Luminal A | 15 (9.9%) | 14(9.3%) | 0.493 |
| | Luminal B(HER2+) | 17 (11.3%) | 19(12.6%) | |
| | Luminal B(HER2-) | 76 (50.3%) | 62(41.1%) | |
| | HER2 overexpression | 24 (15.9%) | 25(16.5%) | |
| | Basal type | 19 (12.6%) | 31(20.5%) | |

**Abbreviations.**
ER, Estrogen Receptor; PR, Progesterone; HER2, Human Epidermal Growth Factor Receptor-2.

## Association between receptor expression heterogeneity and DFS in patients

During a mean follow-up of 54 months (range, 6-205 months), all of the 166 patients (100%) with invasive breast cancer had metastasis, including 43 cases with local metastasis and 123 cases with distant metastasis. Figure 2 shows the Kaplan–Meier curves for all patients with statistically significant differences ($P < 0.05$) between cases with concordant and discordant ER, HER2, and Ki67 between primary and metastatic lesions. For ER expression, the median DFS in primary and metastatic lesions with ER concordant groups (ER(+) →ER(+),ER(-) →ER(-)) and discordant groups (ER(+) →ER(-), ER(-) →ER(+))was 54, 46, 63, and 58 months, respectively(Table 4). Among them, the median DFS of primary HER2(-) to metastatic HER2(+) was 79 months, which was relatively high. Detailed results of median DFS and 95% confidence interval with heterogeneity of EP, PR, HER2, and Ki67 status in primary and metastatic lesions are shown in Table 4.

## Effect of treatment regimen on receptor expression heterogeneity

Among the patients in this article, 166 cases had complete treatment information for ER, 165 for PR, 156 for HER2, and 146 for Ki67. The endocrine therapy was administered to 85 patients in our study, while chemotherapy was given to 160 patients and targeted therapy was provided to 34 patients. The Cox multivariate regression analysis revealed that the expression differences of ER, PR, HER2, and Ki67 were not influenced by chemotherapy and targeted therapy (Table 5). However, a statistically significant difference in ER, PR, and HER2 expression was observed with endocrine therapy.

**Table 3** Descriptive analysis of changes in ER, PR, HER2, Ki67 receptor status, and molecular subtyping of primary and metastatic lesions after excluding unknown data. (n (%)).

| Receptor | Changes in receptor status | | N (%) of changes | Total n (%) |
|---|---|---|---|---|
| ER  (166 cases) | Positive (+) → Negative (−) | | 25(15.1%) | 36(21.7%) |
| | Negative (−) → Positive (+) | | 11(6.6%) | |
| PR  (165 cases) | Positive (+) → Negative (−) | | 41(24.8%) | 53(32.1%) |
| | Negative (−) → Positive (+) | | 12(7.3%) | |
| HER2  (156 cases) | Positive (+) → Negative (−) | | 5(3.2%) | 13(8.3%) |
| | Negative (−) → Positive (+) | | 8(5.1%) | |
| Ki67 (146 cases) | High expression → Low expression | | 24(16.4%) | 52(35.6%) |
| | Low expression → High expression | | 28(19.2%) | |
| Molecular subtyping (151 cases) | Luminal A(15 cases) | →Luminal B(HER2-) | 7(4.6%) | 55(36.5%) |
| | | →HER2 overexpression | 1(0.7%) | |
| | | →Basal type | 3(2.0%) | |
| | Luminal B(HER2+)(17 cases) | →Luminal A | 1(0.7%) | |
| | | →Luminal B(HER2-) | 1(0.7%) | |
| | | →HER2 overexpression | 3(2.0%) | |
| | | →Basal type | 1(0.7%) | |
| | Luminal B(HER2-)(76 cases) | →Luminal A | 9(6.0%) | |
| | | →Luminal B(HER2+) | 3(2.0%) | |
| | | →HER2 overexpression | 2(1.3%) | |
| | | →Basal type | 12(7.9%) | |
| | HER2 overexpression(24 cases) | →Luminal B(HER2+) | 4(2.6%) | |
| | | →Basal type | 2(1.3%) | |
| | Basal type(19 cases) | →Luminal B(HER2+) | 1(0.7%) | |
| | | →Luminal B(HER2-) | 4(2.6%) | |
| | | →HER2 overexpression | 1(0.7%) | |

**Abbreviations.**
ER, Estrogen Receptor; PR, Progesterone; HER2, Human Epidermal Growth Factor Receptor-2.

## Association of tumor size, lymph node metastasis, and receptor expression heterogeneity

A total of 154 and 160 patients had complete T stage (tumor size) and N stage (lymph node metastasis). Research has shown that tumor size is not related to changes in ER, PR, HER2, and Ki67 receptor status ($P = 0.078, 0.680, 0.640, 0.299$). Lymph node metastasis is not associated with changes in ER, PR, HER2, and Ki67 status ($P = 0.631, 0.409, 0.701, 0.918$). The detailed results are shown in Table 6.

## Association between site of metastasis and receptor expression heterogeneity

All 166 patients were followed up for a duration ranging from 6 to 205 months. Metastasis was confirmed through biopsy. Among the patients, 43 patients experienced local recurrence and metastasis in the ipsilateral axillary lymph nodes and chest wall, while distant metastasis occurred in 123 patients. The discordant rates of ER, PR, HER2, and Ki-67 in patients with local recurrence were 25.6%, 20.9%, 7.9%, and 28.6% respectively, whereas those in patients with distant metastasis were 20.3%, 36.1%, 8.5%, and 38.5%

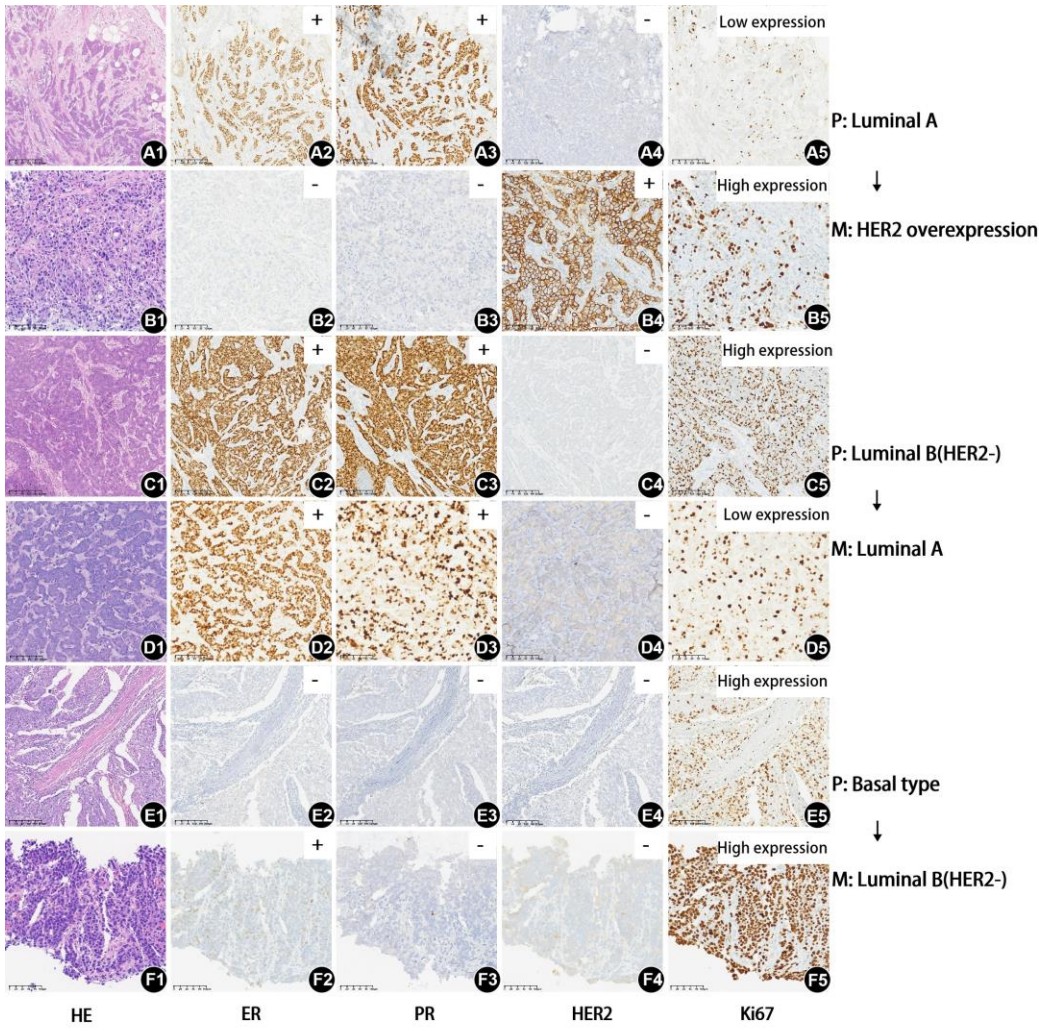

**Figure 1** **HE, ER, PR, HER2, and Ki67 of three cases with inconsistent molecular subtyping in the primary and metastatic lesion.** (A1-A5) The HE, PR, HER2, and Ki67 of the first case's primary lesion, with molecular subtype Luminal A (factor of magnification: 100x) (B1-B5) The HE, PR, HER2, and Ki67 of the first case's metastasis lesion, with molecular subtype changing to HER2 overexpression. (Factor of magnification: 200x) (C1-C5) The HE, PR, HER2, and Ki67 of the second case's primary lesion, with molecular subtype Luminal B(HER2-) (factor of magnification: 100x) (D1-5) The HE, PR, HER2, and Ki67 of the second case's metastasis lesion, with molecular subtype changing to Luminal A (factor of magnification: 200x) (E1-E5) The HE, PR, HER2, and Ki67 of the third case's primary lesion, with molecular subtype Basal type (factor of magnification: 100x) (F1-F5) The HE, PR, HER2, and Ki67 of the third case's metastasis lesion, with molecular subtype changing to Luminal B(HER2-) (factor of magnification: 200x) Abbreviations: P, primary lesions; M, metastatic lesions; ER, estrogen receptor; PR, progesterone; HER2, human epidermal growth factor receptor-2.

respectively. A detailed analysis of the relationship between the site of metastasis and recipient heterogeneity is presented in Table 7.

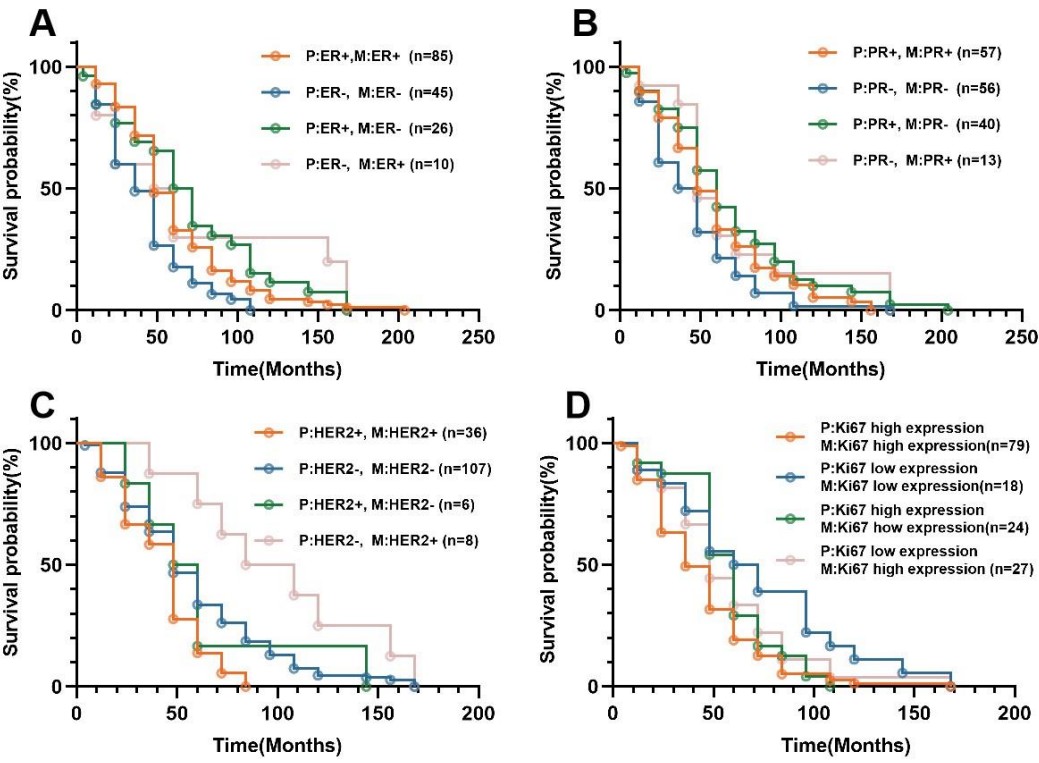

**Figure 2** **Kaplan–Meier survival curves for breast cancer patients with different receptor status at primary and metastatic lesion.** (A) Correlation between ER expression heterogeneity and DFS in 166 patients. (B) Correlation between PR expression heterogeneity and DFS in 165 patients. (C) Correlation between HER2 expression heterogeneity and DFS in 156 patients. (D) Correlation between Ki67 expression heterogeneity and DFS in 146 patients. Abbreviations: P, primary lesion s; M, metastatic lesions; ER, estrogen receptor; PR, progesterone; HER2, human epidermal growth factor receptor-2.

## DISCUSSION

Our study assessed the status of ER, PR, HER2, and Ki67 through immunohistochemical staining of tissue samples obtained from both primary and metastatic lesions in patients with invasive breast cancer. The expression levels of ER, PR, HER2, and Ki67 have significant implications for patient prognosis and treatment outcomes. The comprehensive study demonstrated that discordance between ER and PR status between the primary lesion and metastases in 15%–40% of women and 7%–26% for HER2 status (*Satishkumar, Ramesh & Sanjive, 2023*). The discordance rate in our current study is comparable, and the majority of alterations in ER and PR receptors result in a loss of receptor status. The findings of our study revealed a lack of concordance between primary breast cancer and metastatic breast cancer in terms of ER, PR, HER2, and Ki67 receptors. Specifically, the rates of discordance were 21.7% for ER, 32.1% for PR, and 8.3% for HER2. The majority of studies have demonstrated high conversion rates from positive to negative for ER, PR, and HER2 receptors, as well as from low expression to high expression for Ki67 (*Chamorro et al., 2022*; *Matsumoto et al., 2015*; *Peng et al., 2021*). Conversely, several studies have been

**Table 4** Survival analysis of the median DFS (in months) and 95% confidence interval of patients with different receptor status of primary and metastatic lesions.

| Primary lesion | Metastatic lesion | Median DFS (months) | 95% CI |
|---|---|---|---|
| ER+ | ER+ | 54 | 46.566–61.434 |
| | ER- | 63 | 58.104–67.896 |
| ER- | ER+ | 58 | 20.237–95.763 |
| | ER- | 46 | 35.795–56.205 |
| PR+ | PR+ | 53 | 42.314–63.686 |
| | PR- | 62 | 56.981–67.019 |
| PR- | PR+ | 53 | 29.236–76.764 |
| | PR- | 46 | 34.581–57.419 |
| HER2+ | HER2+ | 47 | 38.768–55.232 |
| | HER2- | 53 | 29.382–76.618 |
| HER2- | HER2+ | 79 | 49.895–108.105 |
| | HER2- | 55 | 48.497–61.503 |
| Ki67 high expression | Ki67 high expression | 42 | 34.416–49.584 |
| | Ki67 low expression | 55 | 50.199–59.801 |
| Ki67 low expression | Ki67 high expression | 50 | 31.850–68.150 |
| | Ki67 low expression | 64 | 35.512–92.488 |

**Abbreviations.**
ER, Estrogen Receptor; PR, Progesterone; HER2, Human Epidermal Growth Factor Receptor-2; DFS, Disease Free Survival; CI, Confidence Interval.

**Table 5** Cox multivariate analysis of prognosis in patients with heterogeneity of receptor expression by treatment regimen.

| Heterogeneous receptors | Characteristic | $\beta$ | $B_{SE}$ | Wald ($\chi^2$) | Exp(B) | 95% CI | P |
|---|---|---|---|---|---|---|---|
| ER (166 cases) | Endocrine therapy | −0.368 | 0.166 | 4.920 | 0.692 | 0.500–0.958 | 0.027 |
| | Chemotherapy | −0.431 | 0.424 | 1.032 | 0.650 | 0.283–1.492 | 0.310 |
| | Targeted Therapy | 0.175 | 0.201 | 0.756 | 1.191 | 0.803–1.768 | 0.385 |
| PR (165 cases) | Endocrine therapy | −0.364 | 0.167 | 4.736 | 0.695 | 0.501–0.965 | 0.030 |
| | Chemotherapy | 0.318 | 0.427 | 0.555 | 0.728 | 0.595–3.174 | 0.456 |
| | Targeted Therapy | −0.191 | 0.201 | 0.902 | 1.211 | 0.557–1.226 | 0.342 |
| HER2 (156 cases) | Endocrine therapy | −0.397 | 0.166 | 5.737 | 0.673 | 0.486–0.930 | 0.017 |
| | Chemotherapy | −0.314 | 0.460 | 0.464 | 0.731 | 0.296–1.802 | 0.496 |
| | Targeted Therapy | 0.318 | 0.210 | 2.299 | 1.374 | 0.911–2.073 | 0.129 |
| Ki67 (146 cases) | Endocrine therapy | −0.341 | 0.176 | 3.764 | 0.711 | 0.504–1.004 | 0.052 |
| | Chemotherapy | −0.180 | 0.428 | 0.177 | 0.835 | 0.361–1.934 | 0.674 |
| | Targeted Therapy | 0.083 | 0.207 | 0.161 | 1.086 | 0.725–1.629 | 0.688 |

**Abbreviations.**
ER, Estrogen Receptor; PR, Progesterone; HER2, Human Epidermal Growth Factor Receptor-2.

conducted high conversion rates from negative to positive for ER, PR, and HER2 receptors, along with a shift from high expression to low expression for Ki67 (*Pizzuti et al., 2021*; *Shen et al., 2020*). In our particular case, we observed a substantial conversion rate from positive to negative for ER and PR receptors, a transition from negative to positive for HER2 receptor status, and an increase in Ki67 expression levels. The treatment recommendations and clinical behavior of metastatic breast cancer will undergo a significant transformation

**Table 6** Chi-square test was used to analyze the association between TN stage and changes in patient receptor status (*n*).

| Characteristic | Receptor Status change | ER | | PR | | HER2 | | Ki67 | |
|---|---|---|---|---|---|---|---|---|---|
| | | YES | NO | YES | NO | YES | NO | YES | NO |
| T (154 cases) | I | 11 | 43 | 19 | 35 | 4 | 48 | 22 | 27 |
| | II | 13 | 63 | 21 | 55 | 6 | 62 | 22 | 45 |
| | III | 6 | 18 | 8 | 15 | 1 | 23 | 5 | 17 |
| | *P* | 0.078 | | 0.680 | | 0.640 | | 0.299 | |
| N (160 cases) | N0 | 4 | 28 | 6 | 26 | 2 | 27 | 9 | 20 |
| | N1 | 14 | 49 | 21 | 42 | 5 | 54 | 21 | 34 |
| | N2 | 10 | 26 | 13 | 23 | 2 | 32 | 11 | 19 |
| | N3 | 7 | 22 | 10 | 18 | 4 | 24 | 9 | 19 |
| | *P* | 0.631 | | 0.409 | | 0.701 | | 0.918 | |

**Abbreviations.**
ER, Estrogen Receptor; PR, Progesterone; HER2, Human Epidermal Growth Factor Receptor-2.

**Table 7** Descriptive analysis of changes in different receptor states at different metastatic sites (*n*(%)).

| Site of metastasis | Cases | Change in ER | Change in PR | Change in HER2 | Change in Ki67 |
|---|---|---|---|---|---|
| **Local recurrence** | 35 | 11 (25.6%) | 9 (20.9%) | 3 (7.9%) | 12 (28.6%) |
| Ipsilateral chest wall | 28 | 9 (25.0%) | 7 (13.2%) | 2 (15.4%) | 10 (19.2%) |
| Ipsilateral axillary lymph nodes | 7 | 2 (5.6%%) | 2 (3.8%) | 1 (7.7%) | 2 (3.8%) |
| **Distant metastasis** | 119 | 25 (20.3%) | 44 (36.1%) | 10 (8.5%) | 40 (38.5%) |
| Contralateral axillary lymph node | 6 | 1 (2.8%) | 3 (5.7%) | 0 (0%) | 2 (3.8%) |
| Lung/pleura | 14 | 2 (5.6%) | 6 (11.3%) | 0 (0%) | 6 (11.5%) |
| Liver | 25 | 5 (13.9%) | 9 (17.0%) | 4 (30.7%) | 7 (13.5%) |
| Bone | 22 | 4 (11.1%) | 9 (17.0%) | 1 (7.7%) | 8 (15.4%) |
| Neck | 12 | 1 (2.8%) | 4 (7.5%) | 1 (7.7%) | 6 (11.5%) |
| Hydrothorax and ascites | 14 | 5 (13.9%) | 5 (9.4%) | 1 (7.7%) | 3 (5.8%) |
| Peripheral lymph nodes | 14 | 4 (11.1%) | 2 (3.8%) | 3 (23.1%) | 5 (9.6%) |
| Others | 12 | 3 (8.2%) | 6 (11.3%) | 0 (0%) | 3 (5.8%) |

**Notes.**
ER, Estrogen Receptor; PR, Progesterone; HER2, Human Epidermal Growth Factor Receptor-2.

as a result of this development. Moreover, the loss of receptor status in metastases can potentially lead to ineffective treatment and adverse drug reactions. *Grinda et al. (2021)* and *Liedtke et al. (2009)* demonstrated that patients with inconsistent receptor performance had poorer prognoses compared to those with consistent receptor expression, possibly due to inappropriate utilization of endocrine therapy and/or targeted therapy. Therefore, it is imperative for recurrent and metastatic breast cancer cases to undergo routine biopsy in clinical practice.

Breast cancer is a highly heterogeneous malignant tumor, characterized by distinct gene expression profiles within the same pathological type of tumor, leading to variations in biological behavior, treatment response, and prognosis evaluation. Consequently, conventional histopathological classification and clinical staging fail to meet the demands for accurate diagnosis and treatment. Integrating different molecular subtypes and prognostic stages can serve as a valuable reference for formulating precise individualized

treatment plans for breast cancer. The rate of discordance by molecular subtype was also assessed, yielding a value of 36.5%. Notably, the transformation rate from Luminal B (HER2-) to basal type exhibited the highest frequency at 7.9%. We have identified 19 cases of basal cell carcinoma, namely three negative breast cancer cases, among which one case has metastasized and converted into Luminal B (HER2+), while four cases have converted to Luminal B (HER2-). Kao et al.'s (*Kao et al., 2021*) study showed a molecular subtyping inconsistency rate of 31.61%, with 10.36% of the molecular subtyping inconsistency group transitioning from receptor positive group (Luminal A, Luminal B, and HER2 overexpression) to Basal type, and 6.74% transitioning from Basal type to receptor positive subtype, similar to our study. In terms of the specific changes in subtype, they investigated statistical difference in survival between patients with concordantly Luminal A phenotype tumor (better outcome) and those with tumors which converted from Luminal A type to TNBC at the metastatic site (poor outcome) (*McAnena et al., 2018*). The diverse molecular classifications result in significant differences in treatment approaches and prognosis. Overall, patients with Luminal type exhibit better prognosis compared to those with HER-2 expression type and basal cell type. The presence of metastases in the molecular classification alteration significantly impacts the prognosis and necessitates a modification in the treatment plan for patients.

The occurrence of local recurrence and distant metastasis is observed in approximately 20%–30% of breast cancer patients following initial treatment. The discordance rates of ER, PR, HER2, and Ki67 in local recurrence were 25.6%, 20.9%, 7.9%, and 28.6% respectively, whereas in distant metastasis they were 20.3%, 36.1%, 8.5%, and 38.5% respectively. The heterogeneity and mechanism of ER, PR, HER2, and Ki67 receptor expression in local recurrent and metastatic lesions remain poorly understood. The inconsistent expression rate of ER, PR, HER2, and Ki67 in distant metastasis has been found to be higher compared to that in local recurrence. This disparity may be attributed to the potential formation of local recurrence resulting from the proliferation of primary tumor cells, which is likely to sustain the expression of the original receptors (*Lowery et al., 2012*). The distant metastatic lesions, however, originate from the remote implantation and proliferation of tumor cells through lymphatic or vascular routes and may exhibit distinct receptors compared to the primary lesions. The expression of receptors in metastatic lesions can be enhanced through endocrine, chemotherapy, or targeted therapy. Therefore, active receptor detection should be performed in cases of local recurrence or distant metastasis to prevent the omission of potential treatments such as endocrine therapy, chemotherapy, or targeted therapy. It remains unclear whether different regions contribute differently to receptor transfer and how changes in their state may influence this process.

The present study demonstrated that, among these cases, patients with negative primary lesions and negative metastases exhibited the shortest DFS for ER, which aligns with the findings of previous research studies (*Zhang et al., 2021*). Patients with early-stage breast cancer who receive tamoxifen for a duration of 5 years demonstrate a significant reduction of 50% in the risk of breast cancer recurrence. However, despite receiving prompt adjuvant tamoxifen therapy, approximately 15 to 20% of patients still experience recurrence (*Santinelli et al., 2008*). This phenomenon can be ascribed to the fact that ER

serves as an independent prognostic factor for breast cancer and is closely associated with DFS (*Jung et al., 2021*). The transition from negative to positive ER status indicates the need for endocrine therapy in order to extend patients' lifespan, thus it is crucial to provide them with this treatment opportunity. The findings of this study demonstrated that the median DFS for PR with positive primary and metastatic lesions, negative primary and metastatic lesions, positive primary and negative metastatic lesions, and negative primary and positive metastatic lesions were 53, 46, 62, and 53 months respectively. The involvement of PR in breast development and its association with the occurrence of breast cancer have been demonstrated. In patients with breast cancer, immunohistochemistry analysis of primary and metastatic lesions reveals a modest reduction in ER levels and a significant decline in PR levels following endocrine therapy, leading to complete loss of PR expression in up to half of the tumors as resistance develops. In our study, there was a transition from positive to negative PR expression in 24.8% of primary and metastatic lesions. However, consensus regarding the impact of receptor status on survival has not yet been reached (*Jiaxin et al., 2022*). The expression of PR accurately reflects the efficacy of endocrine therapy. The synthesis of PR relies on the intact ER-PR pathway. When assessing patient prognosis, changes in PR expression should be considered alongside changes in ER expression. The presence of ER-positive and PR-negative metastatic breast cancers is often indicative of a more aggressive disease phenotype and has been associated with decreased overall survival (*Hou, Peng & Li, 2022*). The higher prevalence of PR-negative tumors in metastatic samples compared to primary tumors is therefore not unexpected. The current study demonstrated that the median DFS was 47 months for patients with primary HER2-positive and metastasis-positive tumors, 55 months for those with primary HER2-negative and metastasis-negative tumors, 53 months for individuals with primary HER2-positive and metastasis-negative tumors, and 79 months for patients with primary HER2-negative and metastasis-positive tumors. These differences were found to be statistically significant ($P < 0.05$). Among these cases, patients with HER2-negative primary lesions and HER2-positive metastases exhibited the most prolonged disease-free survival (DFS). However, numerous studies have consistently demonstrated that patients with altered HER2 expression experience a poorer prognosis compared to those with unaltered HER2 expression. This is likely attributed to the fact that HER2-positive patients receiving trastuzumab treatment exhibit a more favorable prognosis in comparison to HER2-negative patients who do not receive trastuzumab (*Azam, Qureshi & Mansoor, 2009*; *Modi et al., 2022*). HER2 plays a crucial role in facilitating the malignant transformation and proliferation of tumors (*Iqbal & Iqbal, 2014*). We also examined whether the disparity in receptor status between the primary and metastatic lesions could be attributed to variations in treatment. In our cases, there was a significant association between changes in HER2 receptor status and targeted therapy ($P < 0.05$). However, the impact of treatment regimen on receptor alterations between primary and metastatic lesions remains controversial and warrants further comprehensive investigation (*Amir et al., 2012*). In our study, it was observed that patients with low Ki-67 expression in both primary and metastatic tumors exhibited the longest disease-free survival (DFS) and the most favorable prognosis. Conversely, those with high Ki-67 expression in both primary and metastatic tumors

demonstrated the shortest DFS and the poorest prognosis. This association can be attributed to the fact that Ki-67 serves as a reliable indicator of tumor cell activity, exhibiting a strong association with malignant tumor occurrence, metastasis, and overall prognosis. Notably, patients displaying elevated levels of Ki-67 expression are more susceptible to relapse and metastasis (*Ge et al., 2015*).

Changes in receptor expression may represent a genuine biological phenomenon or could potentially arise from inconsistent detection methods. In this particular study, there was no observed association between the expression of ER, PR, HER2, and ki67 with lymph node metastasis status and tumor size. The larger size of the primary tumor and its increased aggressiveness may account for this phenomenon. Our follow-up period in this article is 6-205 months, but the time when the patient first discovers the primary and metastatic lesions, as well as the time of seeking medical treatment, is uncertain, which may affect the accuracy of our follow-up time, thereby affecting survival analysis and receptor heterogeneity. Alternatively, changes in receptor status may arise from genetic mutations or clonal selection occurring during tumor progression, intratumor heterogeneity and clonality, or as a result of systemic therapies such as endocrine therapy, chemotherapy, or trastuzumab. Receptor expression heterogeneity may be associated with the assessment of false negatives and false positives. The inconsistency rate of receptor status in immunohistochemical staining is influenced by the timeliness of specimen fixation, specimen processing, and staining methods. The reliability of fine-needle aspiration specimens in detecting ER immunostaining may be inferior to that of biopsy, as indicated by research studies (*Gong et al., 2004*).

## CONCLUSIONS

In conclusion, this study has demonstrated significant differences in the expression of ER, PR, HER2, and Ki67 between primary and metastatic tumors. These findings have important implications for subsequent treatment planning and prognosis evaluation. Re-biopsy and re-testing of metastatic breast cancer should be considered in clinical practice to facilitate more precise treatment. The present study, however, is a retrospective analysis with a shorter follow-up time, smaller sample size, and different testing methods. Therefore, further prospective multicenter studies are still needed in the future.

### Funding
This work was supported by the Beijing Jing Jian Pathology Development Foundation (No.JJDYSG2023-028). The funders had no role in study design, data collection and analysis, decision to publish, or preparation of the manuscript.

### Grant Disclosures
The following grant information was disclosed by the authors:
The Beijing Jing Jian Pathology Development Foundation: No. JJDYSG2023-028.

## Competing Interests

The authors declare there are no competing interests.

## Author Contributions

- Gaoxiu Qi conceived and designed the experiments, analyzed the data, prepared figures and/or tables, authored or reviewed drafts of the article, and approved the final draft.
- Xin Zhang performed the experiments, prepared figures and/or tables, authored or reviewed drafts of the article, and approved the final draft.
- Xiaoying Gai performed the experiments, analyzed the data, prepared figures and/or tables, and approved the final draft.
- Xiong Yan conceived and designed the experiments, authored or reviewed drafts of the article, and approved the final draft.

## Human Ethics

The following information was supplied relating to ethical approvals (i.e., approving body and any reference numbers):

The Ethics Committee of Qingdao Central Hospital Affiliated to Qingdao University has granted approval for this retrospective study, with the approval number KY202304801.

## Data Availability

The raw data are available in the Supplemental File.

## Supplemental Information

Supplemental information for this article can be found online at http://dx.doi.org/10.7717/peerj.17377#supplemental-information.

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
