# Peer review of "Retrospective analysis of estrogen receptor (ER), progesterone receptor (PR), human epidermal growth factor receptor-2 (HER2), Ki67 changes and their clinical significance between primary breast cancer and metastatic tumors"

_PeerJ, doi:10.7717/peerj.17377_

## Round 0.1 · original submission · Minor Revisions

The manuscript is a retrospective study. The sample is small, but the data are evaluated under various statistical assays to validate the results and conclusion. Nevertheless, it is relevant to actualize and increase the references. This action could increase the quality of the manuscript. Please include in the discussion if the range of following times (4 to 201 months) for patients could affect the results.

·

Basic reporting

The manuscript by Qi and coworkers titled 'Retrospective analysis of ER, PR, HER2, Ki67 changes and their clinical significance between primary breast cancer and metastatic tumors' describes the clinical significance IHC based expression of ER, PR, HER2 and Ki67 in primary breast cancer and metastases. The study was conducted on a small sample size. However, statistical analysis on the limited sample size is rigorous. I have the following comments on this manuscript.
1. Authors should not use the word 'correlation' for analysis not involved in correlation between two continuos variables. I suggest authors to use the words like 'association' or 'mutual exclusivity' instead.
2. Table4 was not cited in the text.
3. Please show IHC staining figures.
4. Please show median survival time with 95% CI and log rank test p value in figure 2.
5. for figures and tables, state which statistical test was used in the analysis in figure legends or foot notes.

Experimental design

None

Validity of the findings

None

Additional comments

None

Reviewer 2 ·

Basic reporting

Basic reporting
Abstract
• To include a clear results statement reporting that patients with discordance experience survival differences and add in data as appropriate.
Introduction
• The introduction requires more detail, including updated literature sources to explain the challenge caused by molecular heterogeneity in metastatic breast cancer.
• Lines 46-47 States: ‘There is increasing evidence indicating that approximately 46 40% of primary and metastatic tumors exhibit altered receptor expression, which necessitates modifications to the treatment plan [2, 3].’ Action: The references for this are 15 years old - Liedtke 2009 and Broom 2009 and require updating. There is a study in a comparable ethnic population (included in discussion) to include: Receptor discordance and phenotype change in metastatic breast cancer - ScienceDirect Kao 2021

Overall the basic reporting is fine in the manuscript article.

Experimental design

Overall the experimental design is fine. The molecular heterogeneity in metastatic breast cancer presents multiple clinical challenges in accurately characterizing and treating the disease and research question is of importance to investigate.

Some specific points to address are:
• What Clinical guidelines were used for treatment management of breast cancer patients?
• Additional information on the types of treatments provided to patients and dosing schedules.
• Line 77 - Follow up the patients: please rephrase
• Statistical test to assess discordance: Kao 2021 used the McNemars test to assess the discordance of ER, PR, HER2 and phenotypes were compared in the primary tumors and metastatic site. Noting that the Kappa consistency test was used in this study, can you provide additional detail on the methodology for Kappa used and please explain why you used this method. Please note findings of this paper: Kappa statistic considerations in evaluating inter-rater reliability between two raters: which, when and context matters | BMC Cancer | Full Text (biomedcentral.com)

No detail provided on specific types of therapies used and doses and guidelines.

I would like to see more detail in the methods used with a few supporting references. Not enough detail to replicate without key references.

Validity of the findings

The rationale and benefit of the study is clearly stated.

Abstract Conclusions: The expression levels of ER, PR, HER2, and Ki-67 in primary and metastatic breast cancer exhibit heterogeneity, which is closely associated with the prognosis and treatment outcomes of patients. To give specific examples of which molecular markers alterations at the metastatic level are related to increased/decreased survival.

• Lines 214- 271 needs to be more succinct and use short paragraphs for understanding.

---

## Round 0.2 · Minor Revisions

Considering that the manuscript has a retrospective view, validating the original data used for analysis is relevant. For this, it is pertinent to show representative histological images to clarify the score and normalization of the analysis.

·

Basic reporting

I thank the authors for their responses.
I am concerned with author's reluctance to show immunohistochemistry images. The reasons they gave for not showing the staining are not sound. If some publications did not show images, it will not be used as a reason for you not showing your data. There is copious amount of literature without scientific rigor. It can't be used as an example. It doesn't matter if your analyses are yin/yang or high/low, all your results were interpreted from IHC staining. if you have done it, show it. I agree that ER, PR and HER2 staining is very common. It seems authors are unaware of the frequent occurrence of false negative staining of these markers even in clinical settings. Therefore, I request authors to include the following information in the revised manuscript.
Please address these concerns and incorporate the changes in text and in figures/tables accordingly.
1. Please show IHC images. Indicate magnification.
2. Explain what internal and external controls were used for IHC staining. Did they repeat the staining when controls failed?
3. Negative staining in this cohort is more than generally observed which may suggest that staining is suboptimal. For example, ER negative patients' percentage above 30 in primary tumors is a strong indication suboptimal staining or issues with fixation time and/or prolonged cold ischemia time etc. In your cohort, ER negative percentage in primary tumors is 33. If this high percentage of ER negative is resulted because of tissue processing, there are chances for all other IHC staining to show false negatives. Therefore, please show and explain how the results you showed are reliable.
4. In continuation with comment 3, what is minimum and maximum cold ischemic time of these specimen?
What is minimum and maximum fixation time of these specimen? These details will help the readers to assess the reasons for possible false negatives in the data.

Experimental design

Important experimental details are missing. Please see basic reporting.

Validity of the findings

It is unclear (based on the data provided) if the findings are reproducible. This can only be evaluated after authors revised the manuscript addressing the concerns raised by this reviewer.

Reviewer 2 ·

Basic reporting

Basic reporting is improved with improved use of references.

Can the authors do a final format check on how the references appear in the manuscript ? Doesn't look consistent and also spaces missing sometimes.

Experimental design

No comments

Validity of the findings

No comments

Additional comments

A good manuscript that adds value to the field, well done to all the authors.

---

## Round 0.3 · Minor Revisions

The modifications made by the authors give insight into the manuscript by making the acquisition of the sample, management, and analysis of the results transparent. However, a minor issue needs attention.
Please modify line 153 by eliminating personal pronouns to maintain the editorial guidelines of the scientific journal.
On lines 155 to 156, a space must be given between the number and cm.

This modification can be made soon. However, the authors must wait for the reviewers' suggestions. Once count on the valuable suggestions of the reviewers, a final decision will be made on the manuscript.

---

## Round 0.4 · accepted · Accept

While in production, please make the following modifications

1-Please modify line 153 by eliminating personal pronouns to maintain the editorial guidelines of the scientific journal.

2- On lines 155 to 156, a space must be given between the number and cm.